# A Guide to Parent-Child fNIRS Hyperscanning Data Processing and Analysis

**DOI:** 10.3390/s21124075

**Published:** 2021-06-13

**Authors:** Trinh Nguyen, Stefanie Hoehl, Pascal Vrtička

**Affiliations:** 1Department of Developmental and Educational Psychology, University of Vienna, 1010 Vienna, Austria; stefanie.hoehl@univie.ac.at; 2Max Planck Institute for Human Cognitive and Brain Sciences, 04103 Leipzig, Germany; 3Department of Psychology, University of Essex, Colchester CO4 3SQ, UK; p.vrticka@essex.ac.uk

**Keywords:** fNIRS, hyperscanning, synchrony

## Abstract

The use of functional near-infrared spectroscopy (fNIRS) hyperscanning during naturalistic interactions in parent–child dyads has substantially advanced our understanding of the neurobiological underpinnings of human social interaction. However, despite the rise of developmental hyperscanning studies over the last years, analysis procedures have not yet been standardized and are often individually developed by each research team. This article offers a guide on parent–child fNIRS hyperscanning data analysis in MATLAB and R. We provide an example dataset of 20 dyads assessed during a cooperative versus individual problem-solving task, with brain signal acquired using 16 channels located over bilateral frontal and temporo-parietal areas. We use MATLAB toolboxes Homer2 and SPM for fNIRS to preprocess the acquired brain signal data and suggest a standardized procedure. Next, we calculate interpersonal neural synchrony between dyads using Wavelet Transform Coherence (WTC) and illustrate how to run a random pair analysis to control for spurious correlations in the signal. We then use RStudio to estimate Generalized Linear Mixed Models (GLMM) to account for the bounded distribution of coherence values for interpersonal neural synchrony analyses. With this guide, we hope to offer advice for future parent–child fNIRS hyperscanning investigations and to enhance replicability within the field.

## 1. Introduction

Children begin to understand themselves and their social surroundings by engaging in embodied social interactions with others. As this process is interpersonal by definition, it should ideally be studied by simultaneously obtaining data from all interaction partners. However, such a second-person social neuroscience approach has only gained momentum in developmental research in recent years [1]. Methodological challenges made it difficult to obtain neural measures from two (or more) individuals at the same time during naturalistic, live interactions [2]. The emergence of functional near-infrared spectroscopy (fNIRS) and particularly the synchronized measurement of brain activity from parents and their children while they interact with each other—so-called hyperscanning—have recently allowed us to take this step. We are now able to investigate the neurobiological underpinnings of socio-cognitive and affective processes underlying these early social interactions and, thereby, deepening our understanding of child development from a second-person social neuroscience perspective.

Children’s social brain development and the associated question of how children come to understand others have long been studied in experiments utilizing pre-recorded and live social stimuli. The obtained results provided important insights into children’s abilities to decipher social information but were devoid of the information’s social context and especially the dynamic and reciprocal nature of real social exchanges. Traditional developmental neuroscience has, thus, investigated the neural correlates of social perception from a third-person perspective, as children were studied individually and mainly in their role as observers [3]. Yet, children are more than merely passive observers during early social interactions; they are embodied agents able to generate more information by their own actions [4]. To truly assess the neurobiological basis of interpersonal social processes, we, thus, need to study parent–child interactions from a second-person perspective [3].

When a caregiver and a child communicate with one another during an embodied social interaction, they typically fluctuate between aligned and misaligned states [5]. Interpersonal alignment, or synchrony, appears to be of great importance for social interaction, not only concerning the matching of behavior and affective states, but also biological and neural rhythms [6]. Interpersonal synchrony is thought to be rewarding to the interactants, as it facilitates mutual prediction, interpersonal coordination, and allostasis—the latter describing the ongoing interpersonal physiological regulation required to meet the environment’s changing demands [6,7,8]. Recent methodological advancements now allow us to study interpersonal synchrony on various levels, but especially the dynamic, reciprocal alignment of oscillatory brain activity from second-person social neuroscience. The simultaneous brain signal acquisition in a multi-person experimental design has been coined with the term ‘hyperscanning’.

Previous research with adults using hyperscanning provided evidence for interpersonal neural synchronization during verbal and non-verbal communication as well as interpersonal coordination [9,10,11,12]. The underlying mechanistic idea is that when the individual oscillatory brain activities become aligned during social interaction, information can be exchanged in an optimal manner. Accordingly, interpersonal neural synchrony was identified in brain networks related to mutual attention, affect attunement, mentalizing, as well as shared intentions [2,13,14]. The correlational evidence was recently extended by work using multi-brain stimulation to increase interpersonal neural synchrony and subsequent behavioral coordination [15]. These findings provide a causal framework for the social effects of interpersonal neural synchrony.

While the number of adult hyperscanning studies has been steadily increasing, only a few developmental hyperscanning studies—using near-infrared sensors specifically—exist to date. fNIRS uses near-infrared light to indirectly and non-invasively assess neural activity in concentration changes of blood oxygenation levels. Near-infrared light is emitted by sensors and detected by detectors (optodes), which are secured to a participant’s scalp. A certain number of channels in the probe sets index activity occurring in the outermost layers of the cortex located below. Similar to functional magnetic resonance imaging (fMRI), fNIRS measures the hemodynamic response and assumes neurovascular coupling, i.e., increases in blood oxygenation levels upon neural activation and vice versa [16]. Beyond the blood-oxygenation-level-dependent (BOLD) response measured with fMRI, fNIRS obtains three relative concentrations measures: oxygenated hemoglobin (HbO), deoxygenated hemoglobin (HbR), and total hemoglobin (Hb). Typically, when neural activity increases, HbO and Hb increase as well, while HbR concentration slightly decreases. fNIRS is especially suitable to naturalistic, developmental hyperscanning research due to its tolerance of motion and applicability [17]. The few studies using fNIRS hyperscanning in a developmental sample available to date show that parents and children (from preschool to school-age) synchronize their brain activities during cooperative tasks, ranging from standardized button press tasks to more naturalistic interactions, in comparison to various control conditions [18]. Additional studies evidenced interpersonal neural synchrony between adults/parents and children in free play interactions and during emotional video watching [19,20].

Generally, brain signal preprocessing approaches were similar between above-mentioned studies. The overall procedure included data conversion from electrical signals to optical density measures, which were then corrected for motion, automatically or visually checked for signal to noise ratio, spatially filtered, and finally subjected to interpersonal neural synchrony estimations. Interpersonal neural synchrony was either assessed using Wavelet Transform Coherence (WTC) or correlational analyses of the two time-series (e.g., Pearson, robust and cross-correlations) [21,22,23]. Here, we will primarily focus on WTC to assess the relation between the two time-series of interacting partners’ brain activity. WTC is suggested to be more suitable in comparison to correlational approaches, as it is invariant to interregional differences in the hemodynamic response function (HRF) [24]. Correlations, on the other hand, are sensitive to the shape of the HRF, which is assumed to be different between individuals (especially regarding age) as well as different within distinct brain areas. Moreover, a high correlation may be observed among regions that have no blood flow fluctuations.

In the remaining parts of this paper, we outline the data analysis procedure we have developed to study fNIRS hyperscanning data from naturalistic parent–child interactions. We provide insight into our approach by providing an exemplary dataset of 20 mother–child dyads during a naturalistic, cooperative versus individual problem-solving task to showcase all pre-processing and analysis steps. Statistical analyses will allow preliminary inferences on whether parent–child dyads show significant interpersonal neural synchrony, and if so within which region of interest (ROI) and during which experimental condition. We will be using MATLAB and RStudio to pre-process and analyze the data. Interested readers can follow the steps of the described analyses by examining the associated files on OSF (https://osf.io/wspz4/, accessed on 12 June 2021).

## 2. Materials and Methods

### 2.1. Sample Description

Our exemplary dataset consists of 20 randomly sampled mother–preschool child dyads from a real study data set and is only used for illustration purposes here. The children were 5–6 years of age (M = 5 years; 3 months; SD = 1.5 months). Mothers’ age averaged at 37.20 years (SD = 3.51 years). Families were recruited from a database of volunteers based in and around a mid-sized city in eastern Germany. All dyads were of European white origin and came from middle to upper-class families based on parental education and family monthly income. Sixty percent of mothers had a university degree and 60% of families had a monthly income higher than 3000 €. Participants were remunerated for their participation.

### 2.2. Experimental Procedure

The primary interest of the study was whether interpersonal neural synchrony between mother and child increases during a cooperative problem-solving task in comparison to an individual problem-solving task. In the cooperative problem-solving task, mother and child sat face-to-face and were instructed to take turns forming seven geometric shapes into predetermined templates (e.g., rocket, bridge, lamp etc.) using wooden blocks (i.e., Tangrams). In the individual condition, mother and child were given four of the same templates and asked to reconstruct these by themselves. Because they were seated at the same table, a portable wooden barrier was put in between them to induce a separated task context. Mother and child were also instructed to refrain from talking to one another if possible. Each task lasted 120 s and was repeated twice, resulting in four task phases (see Figure 1A). In between those four task phases, three 80-s resting phases were included. fNIRS was simultaneously measured in both participants in bilateral temporo-parietal junction (left hemisphere: Channel 9–12; right hemisphere: Channel 13–16) and dorsolateral prefrontal cortex (left hemisphere: Channel 1–4; right hemisphere: Channel 5–8; see Figure 1B). For a full description of the study design, please refer to [25,26].

### 2.3. General Information on fNIRS Data Acquisition and Analysis

A visual summary of all processing steps is depicted in Figure 2. fNIRS data were obtained with a NIRScout 16–16 system (NIRx GmbH, Berlin, Germany) offering 16 sources and 16 detectors, which were divided to measure two interactants. In other settings, two or more devices can be synchronized to enable hyperscanning. The absorption of near-infrared light was measured at the wavelengths of 760 and 850 mm and the sampling frequency was 7.81 Hz. The start of each condition was indicated via triggers sent through an experimental program, for instance, OpenSesame [27]. Varying task conditions can be indicated by triggering a second pin for the end of the condition, while preset and standardized task duration can be manually added at a later time-point.

The NIRStar recording program that came with the NIRScout device used to acquire the present data saved it in a NIRx-specific format—i.e., each participant’s data was stored in a separate folder and included the following files: **.avg*, **.dat*, **.evt*, **.hdr*, **.inf*, **.set*, **.tpl*, **.wl1*, **.wl2*, **config.txt*, and **probeInfo.mat*. Other devices save fNIRS data in different data formats, but all data formats always include a data file comprising the raw wavelength data, as well as a data file in which triggers and/or the optode configuration are saved.

We will use MATLAB and the following toolboxes to analyze the data:

Homer2 (https://www.nitrc.org/projects/homer2, accessed on 12 June 2021; [28]),

SPM for fNIRS (https://www.nitrc.org/projects/spm_fnirs, accessed on 12 June 2021; [29]), and

Cross wavelet and wavelet coherence (http://grinsted.github.io/wavelet-coherence/, accessed on 12 June 2021; [22]). These toolboxes are all freely available online.

### 2.4. Optode Configuration and Raw Data Conversion

In the first pre-processing step, we must store the individual optode configuration as a Source-Detector (SD) file. The SD file can be prepared using the SDgui as part of the Homer2 toolbox, in which the optode positions can be manually entered according to each study’s optode layout. In our case, having used a NIRx system, all necessary information to do so is available in the **probeInfo.mat* and **config.txt* files located in the corresponding data folder. The probeInfo.mat file is automatically created after each recording and stems from a manual configuration using NIRSite set up before the start of data collection. In NIRSite, sources and detectors can be manually placed on MNI head models or imported from digitized optode coordinates. Optode localization only needs to be saved once—if the same optode layout was used for all participants and individual digitizer information is unavailable—and the thereby created SD file can then be used generically for all participants. In this specific case, the function *createSD file* will automatically convert the optode configuration into an SD file.

Once the SD file is prepared, we can run the *NIRxtoSPM* function to convert the stored data into a MATLAB structure with the **.nirs* format, which comprises all needed information for further processing with Homer2. This can be done using both Homer2 and SPM for fNIRS, and their data conversion functions are freely available online. For other systems than NIRx, these conversion functions should be available from the respective manufacturer (e.g., Hitachi: https://www.nitrc.org/projects/hitachi2nirs, accessed on 12 June 2021, Shimadzu: https://www.nitrc.org/projects/shimadzu2nirs/, accessed on 12 June 2021).

### 2.5. Pre-Processing and Visual Quality Check

We start the pre-processing by loading the **.nirs* data of an individual into the MATLAB workspace. Next, we go through a rough automatic data quality check (initial pruning) using the function *enprunechannels*. We specify the expected range of the data (V)—*dRange* = [0.03 2.5], a signal-to-noise threshold—*SNRthresh* = 10 (the lower, the more conservative; with a range from 1–13), and the range of the inter-optode distance—*SDrange* = [2.5 3.5]. Please be aware that these parameters are different for each fNIRS recording device. The obtained output includes the variable *SD.MeasListAct*, which labels bad channels (per row) with a 0. This information can be used to guide the decision on which channels to exclude later.

Subsequently, we convert the raw wavelength data (illustrated in Figure 3) into raw optical density (OD; illustrated in Figure 4, upper panel) using the function *spm_fnirs_calc_od*. Following the initial pruning and data conversion, we proceed to motion correction. The choice of motion correction has been discussed in various other methodological papers [30,31,32]. We have found the MARA algorithm [32] to provide the most efficient motion correction when applied to both our adult and preschool child fNIRS data. MARA implements a smoothing based on local regression using weighted linear least squares and a 2nd-degree polynomial model. We use the default setting for the length of the moving window to calculate the moving standard deviation (L = 1), the threshold for artifact detection (th = 3), and the parameter that defines how much high-frequency information should be preserved by the removal of the artifacts (alpha = 5). This parameter also corresponds to the length of the LOESS smoothing window. Motion corrected OD data is illustrated in Figure 4, lower panel.

Subsequently, we continue with an additional manual, visual quality check. To do so, we convert the OD data to concentration changes (µmol/L) in oxygenated (HbO) and deoxygenated hemoglobin (HbR) using the function *hmrOD2Conc*, with the partial path length factor set to its default [*6 6*]. We then extract the HbO time-series and plot it for all channels using the wavelet transform function *wt*. Following, we check plots for each channel for their integrity, a clearly visible heart band, and visually observable motion artifacts. Examples for different types of artifacts detected during this manual, visual quality check are shown in Figure 5. As a final result, the number of bad quality channels is identified and saved in an array to be later excluded from further analyses.

Once the quality check is finalized, we go back to the OD data after motion correction and apply a band-pass filter, with low- and high-pass parameters of 0.5 and 0.01, respectively, and using a second-order Butterworth filter with a slope of 12 dB per octave. Spatial filtering at this point makes sure that the data is also usable for individual analyses (GLM) and excludes physiological noise to a certain extent.

In the last step, before interpersonal neural synchrony analyses, we once more convert the filtered OD data to concentration changes in HbO and HbR (µmol; illustrated in Figure 6). This time, we use the *spm_fnirs_calc_hb* function from the SPMforfNIRS toolbox, as it allows for age-dependent modification of the modified Beer–Lambert Law. In the current sample, children’s age averaged at the rounded number of 5, and mother’s age averaged at the rounded number of 36. The age-dependent parameters are automatically calculated in the GUI version of SPMforfNIRS toolbox and result in the following parameters: molar absorption coefficients [1/(mM*cm)] of [1.4033 3.8547; 2.6694 1.8096] for both mother and child; distal path length factor of [5.5067 4.6881] for the child, and [6.4658 5.4036] for the mother.

### 2.6. WTC

To calculate interpersonal neural synchrony, several methods have been used, such as Pearson correlation and Robust Correlations [19,33]. We rely on Wavelet Transform Coherence (WTC), the most commonly used method to estimate interpersonal neural synchrony in fNIRS hyperscanning paradigms [21]. WTC is used to assess the relation between the individual fNIRS time series in each dyad and each channel as a function of frequency and time. We estimate WTC using the cross wavelet and wavelet coherence toolbox [22,23].

Based on earlier literature [25,26,34], visual inspection, and spectral analyses, a frequency band of interest needs to be identified. In previous publications on interpersonal neural synchrony during parent–child problem solving, the frequency band of 0.02–0.10 Hz (corresponding to ~10–50 s) was identified as most task-relevant. Accordingly, the same frequency band is used for the present sample. This frequency band should avoid high- and low-frequency physiological noise such as respiration (~0.2–0.3 Hz) and cardiac pulsation (~1 Hz).

Furthermore, coherence values outside the cone of influence need to be excluded in the WTC analysis due to likely estimation bias. Average neural coherence (i.e., interpersonal neural synchrony) can then be calculated for each channel combination and epoch. Epoch estimations, and, therefore, epoch length, are defined by the physical constraints of time-frequency analyses, i.e., the minimal time needed to estimate an appropriate coherence value for the indicated frequency range. Three oscillations in a certain frequency are required for WTC estimations [22]. For example, epoch lengths for a frequency of 2 s must, therefore, be at least 6 s long. To ensure reliable coherence value estimation, most studies have opted towards averaging coherence values over entire conditions. However, few studies have also calculated neural synchrony in smaller epochs to study how it behaves over time [35,36]. Here, we average coherence over the entire condition due to the broad frequency band and relatively short condition duration. Condition onset time can be extracted from the variable *s* included in the initial **.nirs* files. Condition durations were manually added.

To run the WTC function, we extract each individual’s time-series for each channel and combine that time-series array with another column including the timestamps of each row—and, thus, the information on the data’s sampling rate. The timestamp array, if not already available, can be created using the following formula:*t* = (0:(1/*sampling_rate*):((*size*(*time_series*, 1) − 1)/*sampling_rate*))’

We, then, input the following variables into the function *wtc* ([t, hboSub1(:, i)], [t, hboSub2(:, i)], ‘mcc’,0), which translates into timestamp and fNIRS time-series of HbO of Subject 1, timestamps and fNIRS time-series of HbO of Subject 2, and the count for a Monte Carlo Simulation. Generally used to estimate the significance of coherence values, we set the simulation to zero to increase computational speed. The output variables include *Rsq* (coherence values), *period* (period/frequency band), and *coi* (cone of interest). As we need to calculate coherence for each channel combination separately, the rows comprising the frequency band of interest are extracted once for each dyad and used for all channel combinations. The information in *coi* is used to set values outside of the cone of interest to missing values (NaN). To sum up the estimation, interpersonal neural synchrony is calculated by averaging the variable *Rsq* over a certain frequency of interest (*y*-axis) and task duration (*x*-axis), which results in a coherence value for each channel combination in each dyad.

### 2.7. Control Analysis

There are various approaches for additional control analyses. Here, we describe an approach in which we pair a random mother time-series to children’s time-series and bootstrap the pairing 100 times with replacements. Accordingly, 100 WTC values are computed per channel combination and dyad combination with a fixed child, resulting in 2000 values overall. The thereby obtained WTC values are then averaged over each channel combination and pairing and result in a coherence value for each channel combination and child–caregiver dyad. Each true WTC value, therefore, is compared against a “randomly generated” value.

### 2.8. Statistical Analysis

For statistical analysis, the data is exported in a long data format and imported into RStudio (RStudio Team, 2015). Coherence values are bound by 0 and 1 and, therefore, assume a beta distribution. While a Fisher’s r-to-z transformation would be appropriate and indicated for subsequent analyses with a linear (mixed) model assuming a gaussian data distribution, we use the untransformed coherence values to calculate generalized linear mixed models (GLMM) using the package and function *glmmTMB* [37]. We chose to cluster individual channels into regions of interest to enhance the reliability of interpersonal neural synchrony measures in specific cortical regions, even across minor differences in cap fit and, thus, probe settings [20,26,38]. Post-hoc contrasts are conducted using the function and package *emmeans* and corrected for multiple comparison using Tukey’s Honest Significant Difference [39]. The distributions of residuals are visually inspected for each model. Models are estimated using Maximum Likelihood. Model fit is compared using a Chi-Square Test (likelihood ratio test; [40]).

In the current data set, we chose to focus on the analysis of HbO only, for both brevity and its consistent association with neural synchronization (e.g., [25,36,38]). However, we recommend repeating all analyses for HbR (and ideally Hb total) to derive a comprehensive picture of all possible neural synchronization processes [41].

## 3. Results

### 3.1. Control Analysis

First, we analyze the true pair data in comparison to the random pair data using a GLMM. This initial analysis is to ensure that real coherence values are higher than mere spurious and autocorrelations in the data. WTC is entered as the response variable. Condition (cooperation vs. individual), region of interest (left dlPFC vs. right dlPFC vs. left TPJ vs. right TPJ), and pairing (true vs. random) are added as fixed and interaction effects. Due to the small sample size, only random slopes for conditions and random intercepts for dyads are inserted to allow for model convergence. For a larger sample, additional random slopes can be added to the random effects structure.
*wtc* ~ *condition* + *pairing* + *roi* + *condition*: *pairing* + *condition*: *roi* + *pairing*: roi + condition: pairing: roi + (1 + condition | ID)

The GLMM reveals that including an interaction effect between condition and pairing, as well as a fixed effect for region of interest significantly improves the model’s explained variance, χ^2^(1) = 5.396, *p* = 0.020. Individual fixed effects and further interaction effects show no significant increase in explained variance, *p* > 0.056. The full details on model outputs are described in Table 1.

In a post-hoc analysis, we contrast the conditions in each pairing condition, i.e., true vs. random (see Figure 7). In the true pairings, the cooperation condition has higher estimated means (*emmeans* = −0.744, *SE* = 0.010, 95% CI = [−0.764 −0.724]) than the individual condition (*emmeans* = −0.768, *SE* = 0.009, 95% CI = [−0.786 −0.750]). This difference, however, is not significant, *p* = 0.060. In the random pairings, the cooperation condition shows similar levels in estimated means (*emmeans* = −0.757, *SE* = 0.010, 95% CI = [−0.777 −0.738]) as the individual condition (*emmeans* = −0.746, *SE* = 0.009, 95% CI = [−0.765 −0.728]) and their difference is not significant, *p* = 0.388. The results are averaged over all regions of interest. Next, we also contrast the pairings in each condition. In the cooperation condition, true pairings have higher estimated means than random pairings. The difference, however, is not significant, *p* = 0.225. In the individual condition, true pairings have lower estimated means than random pairings. The difference is significant, *t* = −1.98, *p* = 0.047. The results are averaged over all regions of interest.

In a real and sufficiently powered data set, only those channels/averaged regions of interest where there is a significant difference between real and random pairings should be taken forward for interpersonal neural synchrony analyses.

### 3.2. Interpersonal Neural Synchrony

Next, we analyze whether there are condition and regional differences in interpersonal neural synchrony in mother-child dyads. We, therefore, estimate a second GLMM with the true WTC values only. Again, WTC is entered as the response variable. Conditions and regions of interest are entered as fixed and interaction effects. Additionally, we assume random slopes for conditions and random intercepts for each dyad.
*wtc* ~ *condition* + *roi* + *condition* : *roi* + (1 + *condition* | *ID*)

The GLMM reveals that none of the included fixed and interaction effects between condition and region of interest significantly improve the model’s explained variance, *p* > 0.130 (see Figure 8 and Table 2 for model outputs).

## 4. Discussion

### 4.1. Exemplary Dataset

In the current exemplary dataset, we compared interpersonal neural synchrony in frontal and temporal brain regions in mother–child dyads during a cooperative versus individual problem-solving task.

In an initial analysis step, we compared data across all regions and both conditions from true pairs to data from random pairs. In doing so, we found that only true pairs show a trending increase in interpersonal neural synchrony during the cooperation condition as compared to the individual condition. Interpersonal neural synchrony in true pairings, however, is generally not higher than in random pairings in the current data set. Thus, we are unable to conclusively determine that interpersonal neural synchrony is significantly increased beyond spurious correlations in the signal, common task responses, or arousal [40]. Such initial control analysis is crucial when investigating interpersonal neural synchrony, as the results from this exemplary data set show that even random pairings of mothers’ and children’s fNIRS time-series can show high coherence.

To illustrate how to perform the subsequent interpersonal neural synchrony analyses in data from true pairs, we continued to assess the exemplary data, despite the control analysis not revealing a significant difference between true and random pairs. Accordingly, we investigated the true pairs’ interpersonal neural synchrony data for differences between cooperative versus individual problem-solving, as well as between neural regions of interest (i.e., right versus left dorso-lateral prefrontal and temporo-parietal cortex). In this exemplary data set, however, such analysis did not reveal any significant effects.

### 4.2. General Considerations

In our approach, we chose to assess interpersonal neural synchrony across regions of interest (by clustering data from up to four paired channels) rather than from single channel pairings, in accordance with other literature [20]. It is, however, feasible to analyze and report interpersonal neural synchrony from single channel pairings, if appropriate correction for multiple comparisons is employed [42]. At the same time, we would advise caution when establishing interpersonal neural synchrony measures across all possible channel pairings. This is because when computing coherence between channels with different anatomical locations in the assessed individuals, results can be difficult to interpret without clear a-priori hypotheses. The above said, such an approach can yield meaningful results, for instance, if activity in a speaker’s brain regions associated with speech production is linked with a listener’s activity in regions associated with speech perception [43].

A related and more general consideration pertains to the issue of interpersonal neural synchrony data interpretation and, thus, the understanding of its (causal) implication for social interaction—as recently pointed out elsewhere [12,14,15,44]. Within this context, a theory of embodied mutual prediction was proposed [45], which intrinsically links the emergence of coherent brain signals across two (or more) interacting individuals with behavioral monitoring and action prediction. Along these lines, it is crucial that interpersonal neural synchrony data is assessed and interpreted in tight association with behavioral as well as potentially other physiological data obtained during the social interaction simultaneously with fNIRS hyperscanning data. Such a multi-level approach allows for a more comprehensive account of bio-behavioral synchrony across several behavioral, physiological, and neural systems [6]. Furthermore, what emerges from such considerations is the need for appropriate experimental control conditions, beyond comparing an active and interactive experimental task to a simple resting condition.

Most of the so far available studies on interpersonal neural synchrony—especially pertaining to fNIRS hyperscanning research in parent–child dyads—have focused on establishing that interpersonal neural synchrony is increased during cooperation. Such research has crucially advanced our understanding of the importance of bio-behavioral synchrony for social interaction by extending it from behavior and physiology to brain activation. As pointed out in the introduction, however, parent–child interaction is characterized by a fluctuation between aligned and misaligned states [5]. In other words, it is likely that not only short times of high interpersonal neural synchrony, but more extended periods of de- and re-synchronization are determining the quality and future impact of social interactions for child development (see also [7]). Interpersonal neural synchrony should, therefore, be assessed over longer tasks or compared between different task epochs and probed for changes [35,36].

We also need to better understand differences in interpersonal neural synchrony patterns, such as across distinct brain regions, interaction partners, and age. For example, there is emerging evidence that patterns of interpersonal neural synchrony in parent–child dyads seem to diverge for mothers and fathers, with fathers showing a more anatomically localized synchrony pattern [25]. Mothers, on the other hand, seem to show a more anatomically widespread synchrony pattern in frontal and temporal brain areas [26,36]. Furthermore, in an older sample with parents and their school-aged children, more anatomically localized synchrony was evidenced in a cooperative button-press task, in comparison to an anatomically more widespread synchrony pattern in a competitive condition [38]. Differences in interpersonal neural synchrony within parent–child dyads have also been described in relation to the child’s gender, with boys and girls demonstrating differential interpersonal neural synchrony patterns during cooperation with their mothers [46]. Another avenue for future research is to follow changes in patterns of interpersonal synchrony between parent–child dyads longitudinally and in relation to long-term child development and dyadic outcomes.

### 4.3. Limitations

The present guide is not meant to represent an exhaustive review of previous developmental fNIRS hyperscanning research (please refer to [18] for this purpose), nor a comprehensive overview on the current best practices for fNIRS publications (please refer to [47] for this purpose). We furthermore note the following limitations.

The exemplary data set provided here does not comprise an optode registration process, which is typically acquired using fMRI, 3D digitizers, and/or photogrammetry [48,49,50]. Although probe alignment is controlled to a certain degree by using standardized EEG caps aligned to participants’ nasion and inion, variance in optode location could still cause some difficulties in estimating a reliable synchrony value. To avoid misalignment of channel estimations and, thus, synchrony estimates, we opt for a region of interest approach, meaning that synchrony is statistically assessed over groups of four adjacent homologous channels making up four regions of interest. Therefore, synchrony is derived and interpreted for a broader area of the cortex rather than a single homologous channel pair, similar to EEG hyperscanning [21]. Future investigations using fNIRS hyperscanning would, however, benefit from co-registration of optode localization for analyses, including optical reconstruction.

Early naturalistic interactions are still difficult to investigate with fNIRS, as motion artifacts can remain in the data, despite the use of motion correction algorithms. Especially rapid changes in probe distance and separation from the skin during data acquisition can cause spikes in the recorded signal, which commonly available algorithms cannot fully correct. Therefore, future studies need to take both more infant/child optimized optode settings as well as more flexible motion correction approaches into consideration.

Developmental fNIRS hyperscanning can also be affected by systemic effects, such as confounds through task-related arousal [51]. A potential remedy for this issue could be the concurrent assessment of autonomic nervous system and/or physiological responses in the periphery, but also using additional short channel measurements. Although these additional measurements can make hyperscanning with children even more challenging, advancements in wearable diffuse optical tomography might provide a feasible future solution [52].

How we best estimate interpersonal neural synchrony in parent–child dyads remains debated. Although WTC is emerging as a dominant method in fNIRS hyperscanning (see the methods section and [21] for more information), correlational measures are often employed as well [19,33]. If, however, differences in the hemodynamic response increase with age—due to different rates of synaptogenesis and angiogenesis in the developing brain—we might require a novel analysis approach, for instance, cross-frequency coherence/correlation. Moreover, the non-directional analysis with WTC does not allow for interpretations of directionality (i.e., leader–follower relationships) within interpersonal neural synchrony.

## 5. Conclusions

The goal of this article was to showcase a specific implementation to an existing fNIRS hyperscanning pipeline for parent–child dyads. We provided pre-processing steps, interpersonal neural synchrony estimation approaches, as well as the statistical analyses approach for future studies. We used openly available MATLAB toolboxes to build the pipeline, as well as R packages to analyze the data. By making the approach openly available, we hope to enable improvements in standardizing data processing approaches in the field and to enhance replicability of studies.

## Figures and Tables

**Figure 1 sensors-21-04075-f001:**
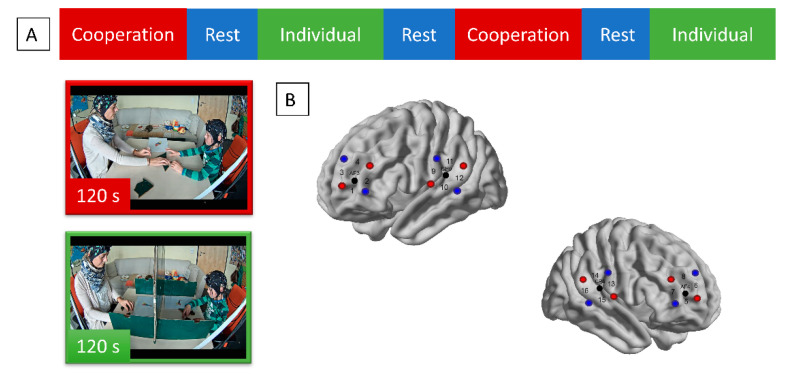
(**A**) Schematic visualization of the experimental procedure and depiction of an example dyad during the cooperation and individual task conditions. Please note that the order of cooperation and individual task conditions were counterbalanced. (**B**) Optode configuration map. fNIRS sources are in red, detectors in blue, channels marked as numbers, and 10–20 EEG landmarks are in black.

**Figure 2 sensors-21-04075-f002:**
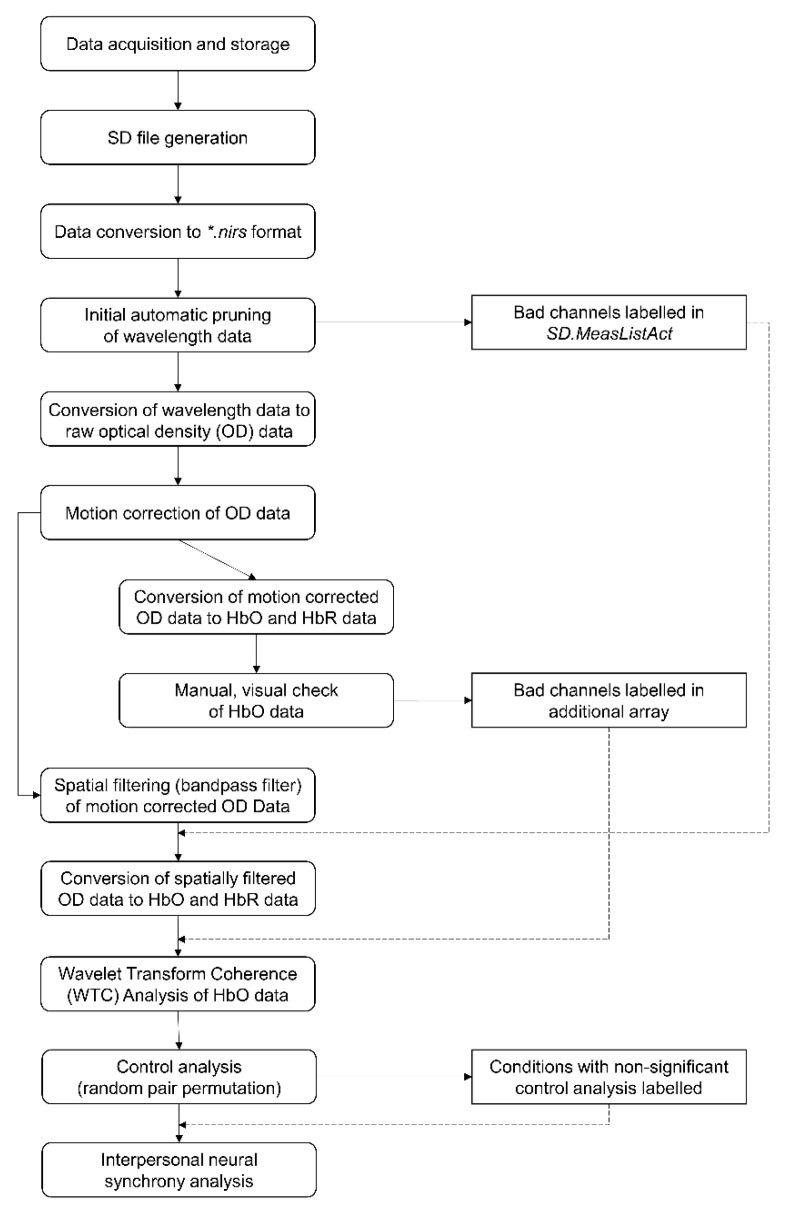
Visual summary of our provided data pre-processing and analysis pipeline.

**Figure 3 sensors-21-04075-f003:**
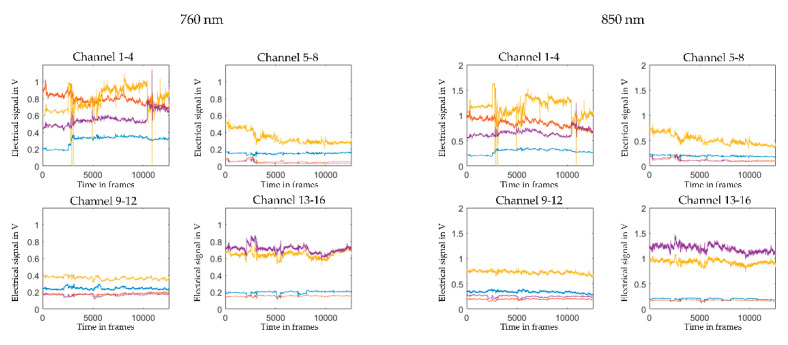
Illustration of raw wavelength data. The graphs depict the raw wavelength data for the two wavelengths 760 nm (**left panel**) and 850 nm (**right panel**) from one individual averaged over the four channels in each region of interest (channel 1–4, left dorsolateral prefrontal cortex; channel 5–8: right dorsolateral prefrontal cortex; channel 9–12: left temporo-parietal junction; channel 13–16; right temporo-parietal junction). The *x*-axis shows the time in frames (sampling rate: 7.81 Hz). The *y*-axis shows optical density units.

**Figure 4 sensors-21-04075-f004:**
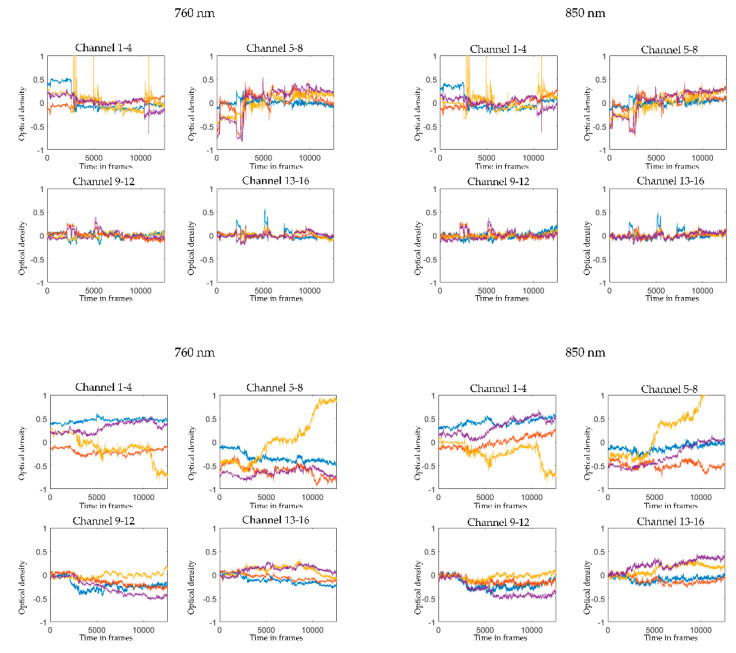
Illustration of raw and motion corrected optical density data. The graphs depict the raw optical density (OD) data (**upper panel**) and the corresponding motion corrected OD data (**lower panel**) in the two wavelengths 760 nm and 850 nm from one individual averaged over the four channels in each region of interest (channel 1–4, left dorsolateral prefrontal cortex; channel 5–8: right dorsolateral prefrontal cortex; channel 9–12: left temporo-parietal junction; channel 13–16; right temporo-parietal junction). The *x*-axis shows the time in frames (sampling rate: 7.81 Hz). The *y*-axis shows optical density units.

**Figure 5 sensors-21-04075-f005:**
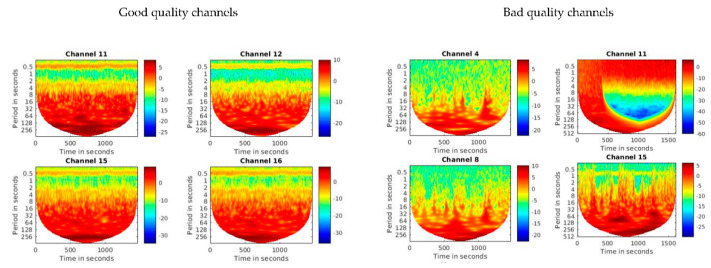
Illustration of good and bad quality channels as determined by the manual, visual data quality check. The graphs depict wavelet transformation plots of the HbO time-series. The *x*-axis shows the time in frames (sampling rate: 7.8125 Hz), the *y*-axis shows the periods in seconds, and the color band depicts the power of the amplitude of the signal (with red indicating highest power). In good quality channels (**left panel**), the heart band (located around period 0.5 s) is clearly visible and continuous and there are no major motion artifacts. In bad quality channels (**right panel**), the heart band is not visible or obscured by strong motion artifacts (channels 4, 8, 15), or there is a very low signal-to-noise ratio (channel 11).

**Figure 6 sensors-21-04075-f006:**
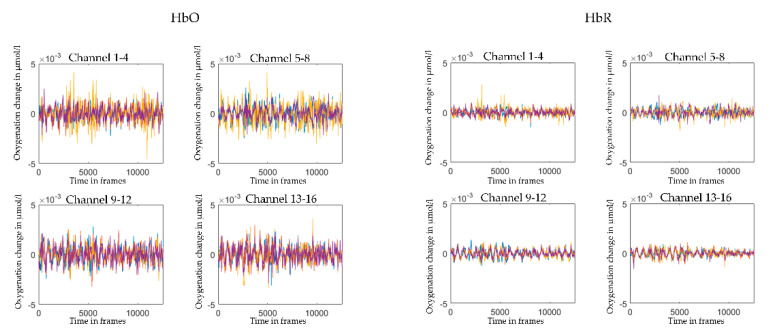
Illustration of concentration changes in HbO and HbR. The graphs depict concentration changes in HbO (**left panel**) and HbR (**right panel**) from one individual averaged over the four channels in each region of interest (channel 1–4, left dorsolateral prefrontal cortex; channel 5–8: right dorsolateral prefrontal cortex; channel 9–12: left temporo-parietal junction; channel 13–16; right temporo-parietal junction). The *x*-axis shows the time in frames (sampling rate: 7.8125 Hz). The *y*-axis depicts the oxygenation change in µmol.

**Figure 7 sensors-21-04075-f007:**
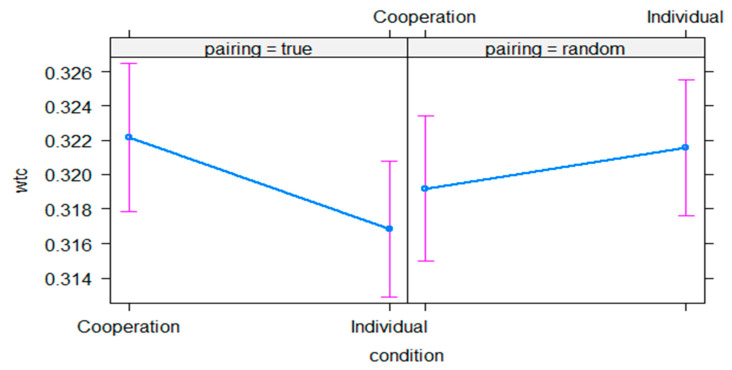
Illustration of true versus random pairing control analysis results. The graph depicts the estimated means of interpersonal neural synchrony (WTC, *y*-axis) in each condition (*x*-axis) divided into true and random pairings (facets). Error bars represent 95 % confidence intervals.

**Figure 8 sensors-21-04075-f008:**
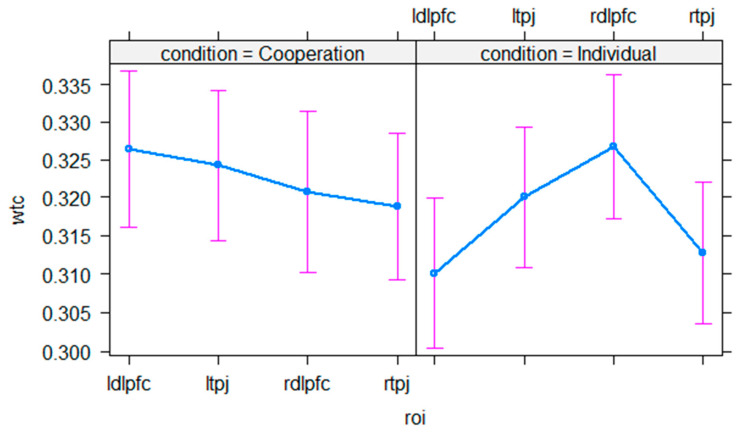
Illustration of interpersonal neural synchrony results. The graph depicts the estimated means of neural synchrony (WTC, *y*-axis) in each region of interest (*x*-axis) divided into the cooperation and individual condition (facets). Error bars represent 95 % confidence intervals.

**Table 1 sensors-21-04075-t001:** Model output for the control analysis.

	Estimates	SE	CI Lower	CI Upper	RE SD	*X²*	*df*	*p*
(*Intercept*)	−0.726	0.018	−0.760	−0.692	0.027			
condition						0.411	1	0.521
*condition individual*	−0.076	0.024	−0.124	−0.030	0.029			
pairing						0.345	1	0.556
*pairing random*	−0.021	0.023	−0.066	0.023				
roi						3.489	3	0.322
*roi ltpj*	−0.011	0.022	−0.055	0.034				
*roi rdlpfc*	−0.025	0.024	−0.071	0.022				
*roi rtpj*	−0.036	0.022	−0.080	0.008				
condition : pairing						5.396	1	0.020
*condition individual* : *pairing random*	0.066	0.032	0.004	0.128				
condition : roi						7.559	3	0.056
*condition individual* : *roi ltpj*	0.057	0.032	−0.005	0.119				
*condition individual* : *roi rdlpfc*	0.102	0.033	0.038	0.165				
*condition individual* : *roi ltpj*	0.048	0.032	−0.013	0.110				
pairing : roi						4.474	3	0.215
*pairing random* : *roi ltpj*	−0.002	0.031	−0.064	0.059				
*pairing random* : *roi rdlpfc*	0.014	0.032	−0.050	0.077				
*pairing random* : *roi rtpj*	0.021	0.031	−0.041	0.082				
condition : pairing : roi						3.959	3	0.266
*condition individual* : *pairing random* : *roi ltpj*	−0.027	0.044	−0.114	0.059				
*condition individual* : *pairing random* : *roi rdlpfc*	−0.082	0.045	−0.171	0.006				
*condition individual* : *pairing random*: *roi rtpj*	−0.012	0.044	−0.098	0.075				

Note. (1) The factor condition had the cooperation condition as reference level. Estimates for the single predictors indicate the change from the response when the predictor changes from the reference level to the level of the predictor (in parentheses). (2) Confidence intervals were derived using the Wald method using the function confint. Included are estimates, standard errors (SE), confidence intervals (CI), the standard deviation of the random effect (RE.SD), and likelihood ratio test outputs for the model comparisons.

**Table 2 sensors-21-04075-t002:** Model output for the interpersonal neural synchrony analysis.

	Estimates	SE	CI Lower	CI Upper	RE SD	*X²*	*df*	*p*
(*Intercept*)	−0.724	0.024	−0.772	−0.677	0.044			
condition						1.540	1	0.214
*condition individual*	−0.075	0.033	−0.140	−0.010	0.053			
roi						4.312	3	0.230
*roi ltpj*	−0.010	0.030	−0.069	0.049				
*roi rtpj*	−0.025	0.031	−0.088	0.036				
*roi rtpj*	−0.034	0.030	−0.093	0.024				
condition : roi						5.647	3	0.130
*condition individual* : *roi ltpj*	0.056	0.042	−0.027	0.139				
*condition individual* : *roi rdlpfc*	0.103	0.043	0.018	0.188				
*condition individual* : *roi rtpj*	0.047	0.042	−0.036	0.130				

Note. (1) The factor condition had the cooperation condition as reference level. Estimates for the single predictors indicate the change from the response when the predictor changes from the reference level to the level of the predictor (in parentheses). (2) Confidence intervals were derived using the Wald method using the function confint. Included are estimates, standard errors (SE), confidence intervals (CI), the standard deviation of the random effect (RE.SD), and likelihood ratio test outputs for the model comparisons.

## Data Availability

Data and analysis scripts supporting reported results can be found on OSF (https://osf.io/wspz4/, accessed on 12 June 2021).

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
