# Peer review of "A Guide to Parent-Child fNIRS Hyperscanning Data Processing and Analysis"

_sensors, 2021, doi:10.3390/s21124075_

Round 1

Reviewer 1 Report

Interesting study.  My only concern is the suitability of the subject matter of this article for this journal.

Author Response

We would like to thank Reviewer 1 for assessing our manuscript. We believe that our submission’s topic is especially well suited for publication in the special issue on Brain Signals Acquisition and Processing in the Journal Sensors – as was pointed out by the special issue editors who personally invited us to contribute. Functional near-infrared spectroscopy (fNIRS) has emerged as a novel and feasible method to study naturalistic social interactions from a second-person social neuroscience perspective by using optical sensors to acquire brain signals in two (or more) people simultaneously. Due to its relative insensitivity to motion and non-invasive nature, fNIRS is the method of choice to study brain activity and interpersonal neural synchrony during parent-child interactions. Because the so far employed data analysis pipelines widely differ between labs, our submission’s aim is to improve consistency and replicability by not only sharing our well-established and validated analysis approach with other groups, but also by providing sample data so that future researchers can use our analysis approach for training purposes.

Reviewer 2 Report

This manuscript presented a study protocol for fNIRS hyperscanning applied in parent-child interaction scenario. It proposed an entire protocol for data collection, analysis and results interpretation. The topic of hyperscanning in fNIRS field is considered to be important. However, several issues including major issues need to be addressed before the manuscript can be considered for publication. Please see my detailed comments below: 

  1. However,  the major issue of the current article is that it did not include a optode registration process. Even an optode map is missing. This makes the article very hard to evaluate. Given fNIRS is an imaging technique that do not provide anatomical information, aligning its probes become critical, especially in the scenario of hyperscanning. Because hyperscanning measures two person at the same time, and the resulting data will be analyzed in pairs. Thus misalignment of the signal collected from different channels is very likely to cause invalid results. The current localization method for fNIRS used fMRI, 3D digitizer, or photogrammetry (see refs below). However, there is yet standard pipeline for localization process for the hyperscanning data analysis. 

Wijeakumar S, Shahani U, Simpson WA, McCulloch DL. Localization of hemodynamic responses to simple visual stimulation: an fNIRS study. Invest Ophthalmol Vis Sci. 2012 Apr 30;53(4):2266-73. doi: 10.1167/iovs.11-8680. PMID: 22427541. 

Hu XS, Wagley N, Rioboo AT, DaSilva A, Kovelman I. Photogrammetry-based stereoscopic optode registration method for functional near-infrared spectroscopy. J Biomed Opt. 2020 Sep;25(9):095001. doi: 10.1117/1.JBO.25.9.095001. PMID: 32880124; PMCID: PMC7463164. 

Lloyd-Fox S., Richards J. E., Blasi A., Murphy D. G. M., Elwell C. E., Johnson M. H. (2014). Coregistering functional near-infrared spectroscopy with underlying cortical areas in infants. Neurophotonics 1:025006. 10.1117/1.NPh.1.2.025006 

  1. It is suggested to add a figure to depict the experimental protocol, as the current description in the experimental procedure is confusing. For example, the statement of "Each task lasted 120 seconds and was repeated twice, with three 80-second resting phases in between." Is confusing, as how three 80 sec rest period can be added between the two task phases. I guess there should be one pre-, one post- and one in-between the two task phases. 
  2. For  the NIRS Scout system, please double confirm the source detector numbers. A NIRScout 8-16 system should have 8 sources and 16 detectors? 
  3. Please provide details about how the probeInfo.mat file was created. Also, as I mentioned above, only having the probeInfo design process may not be enough for the fNIRS hyperscanning application. 
  4. Please provide the units that used in the pre-processing section, e.g. the units for the dRange, and SDrange. 
  5. The authors should also investigate Mayer's wave at 0.1 Hz, and provide details on how did they handle this type of physiological noise. Especially, this frequency overlap with the frequency band of interest (0.06-0.15 Hz) 
  6. Please either provide a justification or reference for the statement of "Three oscillations in a certain frequency are required for WTC estimations." 
  7. The purpose of the statistical analysis was not clear to the reviewer. Are the authors try to analyze the WTC coefficient? Please consider rephrase the section. 
  8. For the statistical results presented in Table 1, have the authors conducted any multiple comparison correction?  
  9. Please consider rearrange Table 1 and 2, as it is very hard to read the first column. 

Reviewer 3 Report

In the paper of Nguyen et al. titled “A Guide to Parent-Child fNIRS Hyperscanning Data Processing and Analysis” the authors provide a guide how to process and analyse fNIRS hyperscanning data with Matlab and R. This is a timely contribution since fNIRS hyperscanning is gaining increasing popularity and there is a need for published guidelines helping to explain the complex data analysis involved. I therefore value the manuscript of Nguyen et al. as a significant contribution.

Comments:

(1) I recommend reading the following new paper …

Best practices for fNIRS publications

MA Yücel, A Lühmann, F Scholkmann, J Gervain, I Dan, H Ayaz, D Boas, ...

Neurophotonics 8 (1), 012101

… and improve the current manuscript based on the several suggestions described in this paper.

(2) Please make the fNIRS data set (raw data) publically available so that readers can download it and can try to replicate the analysis. This would be very helpful for all newcomers in the field.

(3) All figures are blurry and contain compression artifacts. Please include them with a higher resolution. Ideally, create the figures with a vector graphics software program (based on the eps images from Matlab and R) and save then the images as TIF or PNG.

(4)  Fig. 2 + 3 + 5: I recommend improving the x-axis scaling. Limit the x-axis to the time when the experiment ended (not 15000). Remove the white area where not data is shown. Add axis labels too.

(5) Which fNIRS data was used for the analysis? HbO or HbR? This is somehow missing. I recommend to perform the analysis with both data (and ideally also with tHb) since each parameter is affected by physiological noise differentially.

(6) It would be also good to include a statistical model comparison as the end in order to evaluate which model is the best describing the data.

Author Response

Reviewer 3: In the paper of Nguyen et al. titled “A Guide to Parent-Child fNIRS Hyperscanning Data Processing and Analysis” the authors provide a guide how to process and analyse fNIRS hyperscanning data with Matlab and R. This is a timely contribution since fNIRS hyperscanning is gaining increasing popularity and there is a need for published guidelines helping to explain the complex data analysis involved. I therefore value the manuscript of Nguyen et al. as a significant contribution.

>> We thank the reviewer for the positive evaluation.

(1) I recommend reading the following new paper …

Best practices for fNIRS publications

MA Yücel, A Lühmann, F Scholkmann, J Gervain, I Dan, H Ayaz, D Boas, ...

Neurophotonics 8 (1), 012101

… and improve the current manuscript based on the several suggestions described in this paper.

 >> We would like to thank Reviewer 3 for pointing out the recent publication on best practices for fNIRS publications by Yücel et al. We have now included the corresponding reference in the limitation sections to guide readers to a more exhaustive guideline on best practices, as follows:
“The present guide is not meant to represent an exhaustive review of previous fNIRS hyperscanning research (please refer to [18] for this purpose), nor a comprehensive overview on the current best practices for fNIRS publications (please refer to [47] for this purpose).  We furthermore note the following limitations.”

In addition, we added a new figure to detail the experimental procedure as well as the optode configuration and added further information on how the study could be improved (as also requested by Reviewer 2).

“Although probe alignment is controlled to a certain degree by using standardized EEG caps aligned to participants’ nasion and inion, variance in optode location could still cause some difficulties in estimating a reliable synchrony value. To avoid misalignment of channel estimations and thus synchrony estimates, we opt for a region of interest approach, meaning that synchrony is statistically assessed over groups of four adjacent homologous channels making up four regions of interest. Therefore, synchrony is derived and interpreted for a broader area of the cortex rather than a single homologous channel pair, similar to EEG hyperscanning [21]. Future investigations using fNIRS hyperscanning would, however, benefit from co-registration of optode localization for analyses, including optical reconstruction. “

(2) Please make the fNIRS data set (raw data) publicly available so that readers can download it and can try to replicate the analysis. This would be very helpful for all newcomers in the field.

>> Thank you for pointing out the importance of data availability. We uploaded the data to an OSF repository (https://osf.io/wspz4/) and made all raw and processed data as well as MATLAB and R scripts available to interested readers. This is indicated at the end of the introduction and the data availability indications at the end of the paper.

(3) All figures are blurry and contain compression artifacts. Please include them with a higher resolution. Ideally, create the figures with a vector graphics software program (based on the eps images from Matlab and R) and save then the images as TIF or PNG.

>> We are sorry for the seemingly low resolution of image files. As they were saved as PNG with a high resolution, we assume that this was caused by a PDF conversion issue during submission. We  now included all individual figures in the submission portal for better visibility.

(4)  Fig. 2 + 3 + 5: I recommend improving the x-axis scaling. Limit the x-axis to the time when the experiment ended (not 15000). Remove the white area where not data is shown. Add axis labels too.

>> We thank Reviewer 3 for their recommendation. We have now limited all x axis scaling to when the experiment is ending and have added axis labels to all figures.

(5) Which fNIRS data was used for the analysis? HbO or HbR? This is somehow missing. I recommend to perform the analysis with both data (and ideally also with tHb) since each parameter is affected by physiological noise differentially.

>> In the current guide, we only present an exemplary data analysis of HbO. However, as the reviewer has correctly noted, the analysis of both HbO and HbR is highly recommend. Therefore, the following clarification was added to section 2.8:
“In the current data set, we chose to focus on the analysis of HbO only, for both brevity and its consistent association with neural synchronization (e.g., [25, 36, 38]). However, we recommend repeating all analyses for HbR (and ideally Hb total) to derive a comprehensive picture of all possible neural synchronization processes [40].

(6) It would be also good to include a statistical model comparison as the end in order to evaluate which model is the best describing the data.

>> The statistical analyses included model comparisons using a Chi-Square Test. We have added the model formulae to the manuscript to clarify which model provided the best model fit (see section 3.1).

Model 1: wtc ~ condition + pairing + roi + condition : pairing + condition : roi + pairing : roi + condition : pairing : roi + (1 + condition | ID)

Model 2: wtc ~ condition + roi + condition : roi + (1 + condition | ID)

Round 2

Reviewer 2 Report

The authors have fixed issues pointed out by the reviewers and stated limitation of the study, e.g. not having a proper registration process in the manuscript. The quality of the manuscript was improved.

Reviewer 3 Report

I thank the authors for revising the paper. It can now be published.